# Assessment of Hypersensitivity to House Dust Mites in Selected Skin Diseases Using the Basophil Activation Test: A Preliminary Study

**DOI:** 10.3390/medicina60101608

**Published:** 2024-10-01

**Authors:** Magdalena Krupka Olek, Andrzej Bożek, Aleksandra Foks Ciekalska, Alicja Grzanka, Aleksandra Kawczyk-Krupka

**Affiliations:** 1Clinical Department of Internal Diseases, Dermatology and Allergology in Zabrze, Medical University of Silesia, Poniatowskiego 15, 40-055 Katowice, Poland; magda.krupka94@gmail.com (M.K.O.); aleksandra.foks7@gmail.com (A.F.C.); agrzanka@sum.edu.pl (A.G.); 2Department of Internal Diseases, Angiology and Physical Medicine, Medical in Bytom, Medical University of Silesia, Poniatowskiego 15, 40-055 Katowice, Poland; akawczyk@gmail.com

**Keywords:** IgE, BAT, atopic dermatitis, hand eczema

## Abstract

*Background and Objectives*: Allergy to dust mites (HDMs) plays an important role in atopic dermatitis (AD). However, the role of this allergy in other dermatoses is little known. The aim of this study was to assess hypersensitivity to HDMs in patients with AD or hand disease using the basophil activation test. *Material and Methods*: A total of 52 patients with AD, 57 with hand eczema disease, and 68 healthy volunteers qualified for this study. Diagnosis was based on the Hanifin and Rajka criteria, dermatological assessment, and exclusion of other dermatoses. The participants underwent skin prick tests (SPTs), a basophil activation test (BAT) with *D. pteronyssinus* allergen extract, and the concentration of specific IgE (sIgE) for the same allergen in blood serum was determined. *Results*: Positive results in all tests (SPT, sIgE, and BAT) were obtained (24 (46.2%) patients with AD, 9 (15.8%) with hand disease, and none in the control group for *p* < 0.05). The results of the SPT, sIgE, and BAT correlated with each other in the AD and hand eczema groups (Spearmen correlation test, r = 0.72 or 0.85, *p* < 0.05). However, the BAT was positive more often than the SPT and sIgE for *D. pteronyssinus. Conclusions*: House dust mite hypersensitivity is common in patients with AD and eczema. The BAT may be more sensitive for assessing sensitization to house dust mites, especially in patients with hand eczema.

## 1. Introduction

House dust mites (HDMs) are one of the most important indoor sources of allergic diseases such as allergic rhinitis, allergic asthma, and atopic dermatitis [1]. HDMs are microscopic arachnids which live in every home environment. In temperate climate zones, 60–90% of house dust acarofauna are mites from the Pyroglyphidae family. In European countries, the dominant species is *Dermatophagoides pteronyssinus.* However, there are also *Dermatophagoides farinae, Euroglyphus maynei, Acarus siro, Tyrophagus putrescentiae*, and others. Dust mites are a source of many sensitizing proteins. Extracts of mite feces are rich in allergens and extracts from their cleaned bodies. Dust mites contain about 20 allergen groups. The most important allergen of house dust is the Der p 1 antigen, the content of which in dust ranges from 100 to 100,000 ng per 1 g of clean dust. Der p l from the mite *D. pteronyssinus* is considered the most important allergenic antigen among all mite antigens. It is believed that 75% of all IgE antibodies are produced under the influence of this antigen [1,2]. As a result of exposure to mite allergens, allergic inflammation develops in sensitized individuals. Sensitivity to mites can manifest clinically as atopic asthma, perennial allergic rhinitis, or allergic conjunctivitis and also plays a significant role in AD [3]. HDMs trigger an inflammatory allergic reaction by inducing the production of specific IgE antibodies for the HDM allergen and the recruitment of immunocompetent cells, which cause structural and inflammatory changes in the skin and also in the mucous membrane of the respiratory tract and conjunctiva. Most often, HDM allergy has been considered solely as a Th2-type immune response induced by IgE. During patient exposure to mite allergens, the pattern recognition receptors (PRRs) of innate immune cells first recognize pathogen-associated molecular patterns (PAMPs). Consequently, cytokines and chemokines are released. This leads to the recruitment of other inflammatory cells and proinflammatory cytokines to enhance the inflammatory response [4,5]. However, the pathogenesis of AD seems to be more complex and has not been completely investigated. In AD, both immunological factors play a role (I and IV mechanisms of allergic reactions, according to the Gell and Coombs classification, and non-allergic factors) [6]. AD is a polygenically inherited disease, and only a predisposition to the occurrence of atopy is inherited and modified in the individual’s life by environmental factors [1,6,7]. Many chromosomal loci associated with atopy have been found and characterized in studies. The genes responsible for developing atopy encode cellular receptors, immune chains, cytokines, and transcription factors. In people suffering from AD, polymorphism of these genes is found, which can significantly change the quality and intensity of the immune response. As mentioned earlier, in people suffering from AD, there is a predisposition to developing IgE-dependent hypersensitivity to extrinsic and intrinsic antigens in both mechanisms I and IV of the immune reaction. One of the characteristic phenomena is the preference for differentiation of CD4 lymphocytes toward the Th2 line with impaired proliferation of the Th1 line [6,7,8]. The imbalance in the proportions between lymphocyte populations is explained by the weaker local immunity to bacterial and viral infections, the tendency to colonize the skin by microbial pathogens, and reduced and delayed hypersensitivity in people with AD. The consequence of the increased activity of Th2 lymphocytes is the increased production of cytokines released by these cells, primarily Il-4, 5, and 13, and at the same time, reduced synthesis of IFN-g. In patients with AD, there is a functional insufficiency of the epidermal barrier, and therefore allergens can easily penetrate from the skin surface to its deeper layers, and there, with the help of antigen-presenting cells, they stimulate sensitized memory lymphocytes [1,6]. Another important cause is the epidermal barrier defect found in patients with AD, which is genetically determined and consists of many abnormalities in the composition and function of the epidermis. The clinical consequences of this condition are excessive trans-epidermal water loss, dry skin, a reduced itching threshold, high sensitivity to non-specific irritants, the tendency to develop inflammation, and increased penetration of potential allergens and irritants. Finally, environmental factors such as climatic conditions, environmental pollution, and food and inhalant allergens should be emphasized [7,8].

The role of allergy HDMs in different dermatoses is still interesting, such as in hand eczema, where it is often impossible to determine the factor causing the disease. Hand eczema is an often complex disease, many times having an unknown cause. Chronic hand eczema is defined as a persistent, noninfectious inflammatory condition of the skin of the hands. The annual incidence of the disease is estimated to be as high as 10%, and the average duration of lesions is more than 10 years. The clinical diagnosis of hand eczema encompasses a heterogeneous etiological group of diseases, but its terminology lacks a universally accepted classification. Diagnosis was based on etiology when possible and on lesion morphology when necessary. Diepgen et al. presented a classification by etiology which distinguished atopic hand eczema, allergic contact dermatitis (ACD), and irritant contact dermatitis (ICD) both in isolated and combined forms and hand eczema, composed of atopic hand eczema and ICD. In cases without obvious causative or etiological factors, hand eczema was classified according to the morphology of the lesions [9]. Many factors influence the appearance of hand eczema: genes, individual skin structure, environmental factors, lifestyle, as well as food and contact allergens [10,11]. Unlike the described atopic dermatitis, sensitivity to HDMs in these patients is not obvious and is rarely confirmed by IgE tests, among others. On the other hand, many patients reported that their symptoms worsened when exposed to dust, just like patients with atopic dermatitis. We do not have clear evidence on whether HDMs can act as a contact allergen or whether they stimulate systemic allergic inflammation [10,11].

The aim of our study was to assess hypersensitivity to house dust mites in patients with AD or hand eczema using different diagnostic methods: the basophil activation test, skin prick test, and specific IgE for *D. pteronyssinus.*

## 2. Material and Methods

### 2.1. Patients

In all, 52 patients with a final diagnosis of AD, 57 patients with hand eczema, and 68 control healthy volunteers were included in the observation. The inclusion criteria were as follows:

1. Age between 18 and 65 years.

2. Diagnosis of AD confirmed based on the Hanifin and Rajka criteria, medical documentation, and therapy for a minimum 12 months or hand eczema confirmed based on history and physical examination without other skin changes (no skin lesions in other skin regions, low IgE value, and no Hanifina and Rajka criteria).

Patients diagnosed with hand eczema had symptoms for at least a year. All patients had negative contact test results (according to the European standards for contact allergens), had negative skin prick tests with inhalant and food allergens, and did not have bacterial or fungal infections (negative microbiological tests of skin lesions). The association of hand eczema with occupational exposure was also excluded. The preliminary diagnosis was atopic or irritant eczema.

3. Informed consent was given.

The exclusion criteria were concomitant different dermatoses, such as AD and hand eczema, non-allergic diseases, and systemic chronic disease, which can influence skin dermatoses.

A control group was formed from healthy adult volunteers without dermatoses and IgE-mediated clinical allergies as allergic rhinitis, asthma, or food allergy based on their medical history.

The characteristics of the group are presented in Table 1.

### 2.2. Procedures

#### 2.2.1. Skin Prick Tests

The SPT was carried out with inhalant allergens (HAL Allergy B.V, Leiden, The Netherlands) from the following list: *D. pteronyssinus, D. farinae,* mixed grasses, mixed trees, Alternaria, Cladosporium, mugwort, cat and dog allergens, histamine and a negative control. An HDM allergy was defined as a positive skin test for *D. pteronyssinus* and *D. farinae* with a wheel diameter of at least 3 mm and greater than that for the negative control.

#### 2.2.2. Skin Contact Tests

All patients were tested with the 2019 European baseline series (Chemotechnique diagnosis) with the use of IQ ultra chambers. Readings were performed at 48 h, 72 h, and 96 h. The results were interpreted based on the International Contact Dermatitis Research Group’s recommendations.

#### 2.2.3. sIgE Assay

The serum-specific IgE levels for extracts of *D. pteronyssinus, D. farinae*, rDer p 1, and rDer p 2 were determined with ImmunoCAP (ThermoFisher Scientific, Uppsala, Sweden), following the manufacturer’s instructions. Values are expressed in kU/L. The results were considered positive when the sIgE concentration was greater than 0.35 IU/mL (according to the manufacturer’s instructions).

### 2.3. Basophil Activation Test (BAT)

The BASOTEST kit (Glycotope Biotechnology GmbH, Heidelberg, Germany) was performed using the manufacturer’s manual, which called for venous blood sampling. In brief, 100 mL of blood was pre-incubated for 10 min at 37.5 °C in a water bath with 25 mL of reagent. In the second phase, the cells were mixed with 100 mL of the negative control, 100 mL of the positive control, and 100 mL of a commercially available extract mix of *D. pteronyssinus* diluted at 1 μg/mL. The degranulation process was stopped by chilling on ice for 5 min. Then, 25 mL of reagent F, including PE-conjugated anti-IgE as well as FITC-conjugated anti-CD63mAb, was added and further incubated for 25 min. Flow cytometric analysis was performed with the FACScan system (BD, Heidelberg, Germany). The basophils were identified as IgEþ and activated basophils as CD63-FIT, as shown in Table 2.

In each assay, 300–500 basophils were assessed. The upregulation of the activation marker CD63 was assessed by the percentage of the CD63-positive cells compared with the total number of basophilic cells.

### 2.4. Statistical Analysis

Statistical analysis was carried out using Statistica 8.11 (SaftPOl, Cracow, Poland). An ANOVA test, Wilcoxon test, and Student’s *t*-test were used to compare relevant variables. A Spearman correlation test was used to assess any relation between all allergy tests. A *p* value  < 0.05 was considered significant.

## 3. Results

Hypersensitivity to house dust mites occurred most often in AD patients and significantly less in patients with hand eczema. Positive results in all tests (SPT, sIgE, and BAT) were obtained (24 (46.2%) patients with AD, 9 (15.8%) with hand disease, and none in the control group (*p* < 0.05)). For the results of the skin tests, the specific IgE allergen and BAT correlated with each other in the AD and control groups (Spearmen correlation test, r = 0.72 or 0.85; *p* < 0.05). However, the BAT was positive more often than the SPT and sIgE test for *D. pteronyssinus*. Detailed results are presented in Table 2. An example of a positive BAT for a patient with hand eczema is shown in Figure 1.

The presence of house dust mite sensitization did not correlate with the severity of AD or eczema (Spearman correlation test, r = 0.57; *p* > 0.05). There was also no correlation between age, gender, and HDM sensitization in the study group (*p* > 0.05).

Hypersensitivity to HDMs correlated with concomitant grass pollen allergy and perennial allergic rhinitis in patients with AD (Spearmen correlation test, r = 0.79 or 0.81; *p* < 0.05). However, no associations were observed with other allergens in this group, and no similar associations were observed in the patients with hand eczema.

## 4. Discussion

The importance of sensitivity to house dust mites in the severity of atopic dermatitis has been confirmed in many studies. HDMs may aggravate AD symptoms by either bronchial provocation or direct skin contact [7,12]. There is no precise epidemiological data assessing the occurrence of mite allergies in AD patients. It is generally assumed that patients with this dermatosis have positive tests for inhalant allergens at a rate of 40–60%. Our research documented a similar number of patients with house dust mite sensitivity. The examined patients were not allergic to other inhalant allergens.

The IgE-dependent mechanism against HDMs was confirmed by many investigators, and less documented were delay-type reactions (type IV). In the presented study, mainly the IgE-dependent mechanism of dust allergy was confirmed. However, there were also some positive BATs with negative SPT and IgE results, which may indicate higher sensitivity for this test [4,5].

In patients with eczema, the phenomenon of HDM hypersensitivity is more questionable [11,13]. In some patients, all tests were positive, and this subgroup can be classified as allergic eczema with an IgE-dependent reaction. It should be emphasized that these patients did not meet the Hanifin and Rajka criteria, did not have a high total IgE serum count, and therefore did not have a final diagnosis of AD. Most of them saw a connection with the severity of eczema symptoms during exposure to dust.

The most exciting subgroup was patients with eczema, for whom only positive BAT and negative sPT and sIGE results were obtained. The use of a BAT resulted in the separation of the subgroups with or without positive tests and sIgE for *D. pteronyssinus.* This group was, in terms of percentage, larger than the similar ones for AD. This may prove that the mechanism of hypersensitivity to mites in hand eczema is often not the same as that in AD.

The BAT assesses IgE-dependent mechanisms. The BAT is used in the diagnosis of various drug reactions. The BAT may provide more accurate results than skin prick tests and sIgE tests, such as for food allergies [14,15]. The sensitivity of the BAT’s specificity in hypersensitivity to house dust mites has not been assessed, but for the allergy to Hymenoptera venoms, the sensitivity of the BAT was estimated to be 85–100%, while the specificity was 83–100% [14,15].

In the presented study, more positive BAT results than in the SPT and sIgE test for *D. pteronyssinus* in hand eczema suggest that hypersensitivity to HDMs in this group may be more common than expected. A much smaller discrepancy in the tests performed was observed in the patients with atopic dermatitis, which proves that exposure to dust mites mainly triggers IgE-dependent reactions. In the case of patients with hand eczema, the clinical association observed by patients with skin deterioration after contact with house dust may result from a negative SPT, and sIgE may result from this other immunological reaction. This requires some further research.

This preliminary study had the following limitations: small study groups, limitation to only one mite tested, and an inability to completely exclude other skin diseases. However, research groups will be constantly expanded. *D. pteronyssinus* is the main allergen of HDMs in our geographical area, and therefore we focused on it.

## 5. Conclusions

House dust mite hypersensitivity is common in patients with AD and eczema. The BAT may be more sensitive for assessing sensitization to house dust mites, especially in patients with hand eczema.

## Figures and Tables

**Figure 1 medicina-60-01608-f001:**
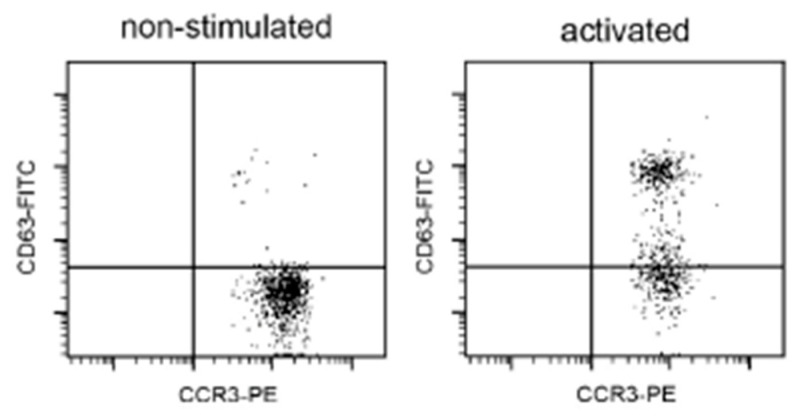
An example patient with atopic dermatitis with a positive BAT result for stimulation with *D. pteronyssinus* extract (1 μg/mL). Legend: CD63- FITC = anti-human CD63-FITC antibody; CCR3-PE = the selection of basophils with CCR3. Basophil activation test (BAT). Percentages of CD63+ cells expressed in response to *D. pteronyssinus* extract in 1 μg/mL dilution.

**Table 1 medicina-60-01608-t001:** The characteristics of the study patients.

Variables	Atopic Dermatitis n = 52	Hand Eczema n = 57	Healthy Controls n = 68	*p*
age: yrs ± SD	36.1 ± 6.9	40.8 ± 9.1 *	33.2 ± 10.3	0.04
women (%)	57.4%	60.1%	55.1%	<0.05
duration of disease: yrs ± SD	12.9 ± 10.1	10.1 ± 4.7	-	<0.05
BMI ± SD	23.6 ± 2.7	26.8 ± 5.7 *	24.1 ± 3.2	0.02
current or former smokers (%)	24.6%	28.1%	26.8%	<0.05
total IgE kU/L ± SD	456.1 ± 106.3 **	59.1 ± 30.3	44.6 ± 28.1	0.001
asthma allergy (%)	27.1% **	4.7%	0	0.004
rhinitis allergy (%)	78.1% **	9.2%	0	0.003
atopy in family (%)	23.7% **	8.1%	4.5%	0.02
polysentization (%)	39.2% **	1.2%	0	0.003
inhalant allergens				
most frequently				
confirmed:				
grass pollen	11.5%	1.5%	0	
trees pollen	7.4%	1.1%	0.5%	
cat	2.3%		0	
food allergy	24.7% **	0	0	0.003

Legend: SD = standard deviation; BMI = body mass index. * Significant differences in group with hand eczema in comparison with others (Student’s t-test or ANOVA test, *p* < 0.05). ** Significant differences in group with atopic dermatitis in comparison with others (Student’s t-test or ANOVA test, *p* < 0.05).

**Table 2 medicina-60-01608-t002:** Results of allergy procedures in studied patients.

Positive Results	Atopic Dermatitis n = 52	Hand Eczema n = 57	Healthy Controls n = 68	*p*
SPT	28 (53.8%)	11 (19.3%)	1 (0.01%)	<0.05
sIgE	32 (61.5%)	6 (10.5%)	1 (0.01%)	<0.05
BAT	34 (65.4%)	14 (24.6)%)	0	<0.05
mean percentage increase of activated CD63 basophils	74.6%	68.1%	2.8%	
correlation r *	0.72	0.85	-	

Legend: SPT = skin prick test with extract of *D. pteronyssinus*; sIgE = specific IgE allergen for *D. pteronyssinus*; BAT = basophil activation test with 1 μg/mL extract of *D. pteronyssinus*. * Spearman correlation test = r.

## Data Availability

The data presented in this study are available on request from the corresponding author.

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
