# Peer review of "Assessment of Hypersensitivity to House Dust Mites in Selected Skin Diseases Using the Basophil Activation Test: A Preliminary Study"

_medicina, 2024, doi:10.3390/medicina60101608_

Round 1
Reviewer 1 Report
Comments and Suggestions for Authors
The manuscript "Hypersensitivity to house dust mites in selected skin diseases using the basophil activation test - preliminary study" focuses on the role of house dust mite (HDM) sensitivity in the development of atopic dermatitis (AD) and hand eczema. The authors justify the relevance of this study by emphasizing the significance of HDMs as a common allergen and the uncertainty regarding their role in hand eczema development. The study involved 119participants, including 52 individuals with AD, 57 with hand eczema, and 68 healthy controls. Skin tests, determination of specific immunoglobulin E (sIgE) levels to D. pteronyssinus, and the basophil activation test were performed. Data analysis included ANOVA, Wilcoxon, Student's t-tests, and Spearman correlations. Resultsshowed that HDM sensitivity was more prevalent amongpatients with AD compared to those with hand eczema.. In the AD group, positive results were obtained in 46.2% of patients in all tests (SPT, sIgE, BAT), while in patients with hand eczema, only 15.8% showed positive results. In the AD group, there was a correlation between SPT, sIgE and BAT, with BAT often being positive even when SPT and sIgE were negative. The authors interpret the data correctly, emphasizing the significance of BAT for diagnosing HDM allergy, particularly in patients with hand eczema. They suggest that the mechanism of HDM hypersensitivity in hand eczema differs from that observed in AD. Theyconclude that sensitivity to HDM is common amongpatients with both AD and eczema and that BAT is a more sensitive method for detecting HDM allergies, especially in those with hand eczema. It should be noted that the study focuses on the important issue of the role of HDM in skin disease development. A clear patient classification is provided, and all research methods used are described. The authors highlight the differences in hypersensitivity mechanisms to HDM between AD and hand eczema. Theycorrectly draw conclusions regarding the role of BAT in diagnosing HDM allergies. However, the small sample size of the study is a limitation. Data obtained from small groups may have limited generalizability. Additionally, there is a lackof information about specific types of hand eczema, assome types may be more closely linked to HDM allergy than others. Furthermore, there is insufficient information aboutconcomitant conditions, such as bronchial asthma or hay fever. Recommendations: It is necessary to increase the sample size in order to increase the statistical power of the study. A more in-depth analysis should be conducted ondifferent types of hand eczema. The study should also include an analysis of other allergens, such as plant pollenand epidermal allergens, and a comparative analysis of patients with different types of allergies. This manuscript presents a prospective study that contributes to ourunderstanding of the role of HDM in AD and hand eczema. Nevertheless, further studies with larger samples are required to confirm the findings.
Author Response
Thank you very much for your valuable review.
Reply for comments : However, the small sample size of the study is a limitation. Data obtained from small groups may have limited generalizability".
Answer: We are aware that the group is small, so we added the information that this is a preliminary study. We will try to expand the group shortly. We also tried to avoid generalizations in the discussion.
"Additionally, there is a lack of information about specific types of hand eczema, assome types may be more closely linked to HDM allergy than others.Furthermore, there is insufficient information about concomitant conditions, such as bronchial asthma or hay fever."
Answer: We added some important information about types of hand eczema (only as clinically preliminary diagnosis before testing), and detials about what kind of eczema was excluded, and also details about concomitant diseases in table.
"The study should also include an analysis of other allergens, such as plant pollen and epidermal allergens, and a comparative analysis of patients with different types of allergies"
Answer: We added some information about hypersensitivity to other allergens (the most frequent positive) and polysenistizations (Table 2).
Thank you
Reviewer 2 Report
Comments and Suggestions for Authors
Authors investigated HDM sensitization, by SPT, specific IgE and BAT with D. pteronyssinus in patients with atopic dermatitis, hand eczema and controls. What they found is that more than 50% of patients with atopic dermatitis are sensitized to HDM, a very well-known information.
Methods
Patients with hand eczema should have been better characterized, by detailed occupational history and patch tests
Control group: It is surprising the absolute lack of sensitization in this group, as HDM sensitization is reported in around 5 to 30 percent of the general population to skin test reactivity.
it is not clear why allergic rhinitis and asthma were considered exclusion criteria; the only exclusion criteria should have been atopic dermatitis and hand eczema.
Discussion
Lines 197-199 Positive BAT test never suggests a type III or type IV hypersensitivity reaction! Positive BAT test depends on the presence of specifi IgE on the basophils’ membrane.
Lines 228-9 Diagnosing hand eczema is only initial step which temporarily excludes other causative factors. Therefore, the subgroup with positive tests was finally diagnosed with allergic eczema. This statement is totally unsound, as Authors evaluated IgE sensitization, which may be responsible for immediate reaction (not eczema, which is a chronic inflammatory response). Patch tests for common aptens should have been performed in patients with hand eczema.
To evaluate a delayed hypersensitivity reaction (i.e. type IV) to HDM the Authors should have done the patch test with HDM.
Comments on the Quality of English Language
minor editing required
Author Response
Thank you very much for valuable review.
Reply for comments :
ethods
Patients with hand eczema should have been better characterized, by detailed occupational history and patch tests:
Answer: PAtch testst were negative with European Standards and also occupational history- these information were added
it is not clear why allergic rhinitis and asthma were considered exclusion criteria; the only exclusion criteria should have been atopic dermatitis and hand eczema.
Answer:
No, some of these patients had allergic asthma and or/allergic rhinitis etc .Date was added in table.
Discussion
Lines 197-199 Positive BAT test never suggests a type III or type IV hypersensitivity reaction! Positive BAT test depends on the presence of specifi IgE on the basophils’ membrane.
Answer. Yes, It was our mistake. This sentence was changed
Lines 228-9 Diagnosing hand eczema is only initial step which temporarily excludes other causative factors. Therefore, the subgroup with positive tests was finally diagnosed with allergic eczema. This statement is totally unsound, as Authors evaluated IgE sensitization, which may be responsible for immediate reaction (not eczema, which is a chronic inflammatory response). Patch tests for common aptens should have been performed in patients with hand eczema.
To evaluate a delayed hypersensitivity reaction (i.e. type IV) to HDM the Authors should have done the patch test with HDM.
Answer We agree with this opinion. Unfortunately, in our study, we did not perform patch tests with house dust mites, but we would like to do such a study in the future. Then, it will be possible to discuss the participation of type IV reactions in more detail. We have removed the unfortunate sentences from the discussion.